# Universal features of higher-form symmetries
# at phase transitions

**Xiao-Chuan Wu[1], Chao-Ming Jian[2] and Cenke Xu[3]**

**1** Department of Physics, University of California, Santa Barbara, CA 93106, USA
**2** Department of Physics, Cornell University, Ithaca, New York 14853, USA
**3** Department of Physics, University of California, Santa Barbara, CA 93106, USA

## Abstract

We investigate the behavior of higher-form symmetries at various quantum phase transitions. We consider discrete 1-form symmetries, which can be either part of the generalized concept "categorical symmetry" (labelled as $\tilde{Z}_N^{(1)}$) introduced recently, or an explicit $Z_N^{(1)}$ 1-form symmetry. We demonstrate that for many quantum phase transitions involving a $Z_N^{(1)}$ or $\tilde{Z}_N^{(1)}$ symmetry, the following expectation value $\langle (\log O_{\mathcal{C}})^2 \rangle$ takes the form $\langle (\log O_{\mathcal{C}})^2 \rangle \sim -\frac{A}{\epsilon}P + b \log P$, where $O_{\mathcal{C}}$ is an operator defined associated with loop $\mathcal{C}$ (or its interior $\mathcal{A}$), which reduces to the Wilson loop operator for cases with an explicit $Z_N^{(1)}$ 1-form symmetry. $P$ is the perimeter of $\mathcal{C}$, and the $b \log P$ term arises from the sharp corners of the loop $\mathcal{C}$, which is consistent with recent numerics on a particular example. $b$ is a universal microscopic-independent number, which in $(2+1)d$ is related to the universal conductivity at the quantum phase transition. $b$ can be computed exactly for certain transitions using the dualities between $(2+1)d$ conformal field theories developed in recent years. We also compute the "strange correlator" of $O_{\mathcal{C}}$: $S_{\mathcal{C}} = \langle 0 | O_{\mathcal{C}} | 1 \rangle / \langle 0 | 1 \rangle$ where $|0\rangle$ and $|1\rangle$ are many-body states with different topological nature.

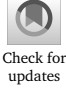
## 1  Introduction

The concept of symmetry is the most fundamental concept in physics, and has profound implications and constraints on physical phenomena. In recent years various generalizations of the concept of symmetry have been explored. For example, ordinary symmetries in a $d-$dimensional system are associated with the global conservation of the symmetry charges, and the symmetry charges localized within a $d-$dimensional subsystem of the space can only change through the Noether current flowing across the surface of the subsystem. In recent years the concept of 1-form symmetry (more generally higher form symmetry) was proposed (see for example Ref. [1–9]), and the concept of 1-form symmetry is associated with conserved "flux" through a $(d-1)-$dimensional subsystem; and the flux can only change through the flowing of a 2-form symmetry current across the edge of the $(d-1)-$dimensional subsystem. The concept of 1-form symmetry was proven highly useful when analyzing gauge fields. Using this new concept of symmetry and its 't Hooft anomaly, it was proven that gauge fields with certain topological term cannot be trivially gapped [10], which is an analogue of the Lieb-Shultz-Mattis theorem in condensed matter systems [11, 12].

Lagrangians are often used to describe a physical system, and the form of the Lagrangian depends on one's choice of "local degrees of freedom" of the system, and other degrees of freedom may become nonlocal topological defects in the Lagrangian. When we select another set of local degrees of freedom of the same system to construct the Lagrangian, it will take a new form, and the new form of Lagrangian is related to the original Lagrangian through a "duality transformation". It was realized in recent years that, in some examples, duality transformation of the Lagrangian, along with the obvious symmetry of the Lagrangian, could be embedded into a larger symmetry group [13, 14], which may only emerge in the infrared limit, and is not explicit unless one takes into account of all the dual forms of the Lagrangian.

Most recently a notion of "categorical symmetry" was developed. For example the $1d$ quantum Ising model has two sets of conservations laws: the conservation of Ising spins, and also conservation of kinks of the Ising spins. The conservation of the Ising spins correspond to an "explicit symmetry" in the Hamiltonian, while the conservation of kinks is governed by an "inexplicit symmetry" in our current manuscript (for further explanation of these notions please refer to the appendix). These two conservation laws can be made both explicit symmetries by embedding the $1d$ system as the boundary of a $2d$ toric code model, and the conservation laws of the Ising spins and kinks arise from the fusion rules of the $e$ and $m$ anyons in the bulk. The notion of categorical symmetry unifies the explicit symmetry of a model and the inexplicit symmetry of its dual model, and treat them on an equal footing [15]. To diagnose the behavior of the categorical symmetries, and most importantly to diagnose the explicit symmetry and the inexplicit dual symmetry on an equal footing, a concept of "order diagnosis operator" (ODO) was introduced, whose expectation value reduces to the correlation function between order parameters for an explicit 0-form symmetry, and reduces to a Wilson loop for an explicit 1-form symmetry [16]. The ODO was also referred to as the "patch operator" in Ref. [15]. For example, the ODO for the $Z_2$ symmetry of the $2d$ quantum Ising model is $O_{ij} = \sigma_i^z \sigma_j^z$, while the ODO for the dual $\tilde{Z}_2^{(1)}$ 1-form symmetry is $\tilde{O}_{\mathcal{C}} = \prod_{j \in \mathcal{A}, \partial \mathcal{A} = \mathcal{C}} \sigma_j^x$, where $\sigma^z$ transforms under

the explicit $Z_2$ symmetry. $\tilde{O}_{\mathcal{C}}$ creates a domain wall of $\sigma^z$ along a closed loop $\mathcal{C}$ by flipping the sign of $\sigma^z$ on a patch $\mathcal{A}$, which is the interior of $\mathcal{C}$ [1]. ODOs for systems with special symmetries such as subsystem symmetries may have special forms and behaviors, and examples with these special symmetries were discussed in Ref. [16].

The expectation value of $O_{ij}$ and $\tilde{O}_{\mathcal{C}}$ in the $2d$ quantum Ising system characterizes different phases of the system. In the two gapped phases, i.e. the ordered and disordered phase of $\sigma^z$, the behavior of $\langle O_{ij}\rangle$ and $\langle \tilde{O}_{\mathcal{C}}\rangle$ are relatively easy to evaluate, since they can be computed through perturbation [18], which is protected by the gap of the phases. In the ordered phase of $\sigma^z$, $\langle O_{ij}\rangle$ saturates to a constant when $|i-j| \to \infty$, and $\langle \tilde{O}_{\mathcal{C}}\rangle$ decays with an area law; in the disordered phase of $\sigma^z$, $\langle O_{ij}\rangle$ decays exponentially with $|i-j|$, while $\langle \tilde{O}_{\mathcal{C}}\rangle$ decays with a perimeter law. But at the critical point of the system, i.e. the $(2+1)d$ quantum Ising phase transition, the behavior of the ODO $\tilde{O}_{\mathcal{C}}$ is more difficult to evaluate. Ref. [19] evaluated $\langle \tilde{O}_{\mathcal{C}}\rangle$ numerically, and the result indicates that in addition to a leading term linear with the perimeter of $\mathcal{C}$, a subleading term which is logarithmic of the perimeter arises for a *rectangular* shaped loop $\mathcal{C}$. The logarithmic subleading contribution may be a universal feature of ODO at a critical point, and the $Z_2$ ODO can be mapped to the 2nd Renyi entanglement entropy of a *free* boson/fermion system [19]. It is known that there is a corner induced logarithmic contribution for the Renyi entropy in a general conformal field theory [20–24]. However, for *interacting* systems the exact relation between entanglement entropy and ODO is not clear yet.

In this work we demonstrate that, for a $2d$ quantum system with either an explicit 1-form symmetry $Z_N^{(1)}$, or an inexplicit symmetry $\tilde{Z}_N^{(1)}$ (which is dual to a 0-form ordinary $Z_N$ symmetry), the following quantity $\langle (\log O_{\mathcal{C}})^2\rangle$ or $\langle (\log \tilde{O}_{\mathcal{C}})^2\rangle$ take a universal form $-\frac{A}{\epsilon}P + b\log P$ at many quantum critical points. Here $P$ is the perimeter of the loop $\mathcal{C}$. $b$ is a universal number which arises from a sharp angle of the loop $\mathcal{C}$; $b$ is proportional to the universal conductivity of the $2d$ quantum critical point, and it is a universal function of the angle $\theta$. We demonstrate this result for various examples of quantum critical points. We also comment on the connection between ODO and entanglement entropy in the end of the manuscript. A logarithmic contribution from angle/cusp of a Wilson loop was found before for $(3+1)d$ gauge field (see for instance Ref. [25]). Our computation is for quantum critical points (QCP) in $(2+1)d$, and the coefficient of the logarithmic contribution is related to a known universal quantity associated to the QCP. Our result is exemplified with multiple concrete examples, the desired quantity of some of the examples can be computed exactly using recently developed duality between $(2+1)d$ QCPs.

We also compute a quantity called the "strange correlator" of the 1-form ODO $O_{\mathcal{C}}$. The strange correlator was introduced as a tool to diagnose the symmetry protected topological (SPT) states based on the bulk wave function instead of the edge states [26], and it was shown to be effective in many examples [27–35]. In the current work we study the strange correlator for one example of 1-form SPT state, but we expect similar studies are worth pursuing for more general cases.

# 2 Systems with dual $\tilde{Z}_N^{(1)}$ 1-form symmetry

## 2.1 Example 1: $Z_N$ order-disorder transition

We first consider cases when the system has an explicit $Z_N$ (0-form) symmetry, and it has an inexplicit dual $\tilde{Z}_N^{(1)}$ 1-form symmetry. The simplest example of quantum phase transition, is the order-disorder transition of the $Z_N$ symmetry. The lattice model with $Z_N$ symmetry, can be

---

[1]For the case of Ising model, the ODO the dual inexplicit symmetry was also called the disorder operator in Ref. [17]. For more explanation please refer to the appendix.

embedded into an ordinary U(1) rotor model:

$$H = \sum_{<i,j>} -t\cos(\hat{\theta}_i - \hat{\theta}_j) + V(\hat{n}_i) - 2u\cos(N\hat{\theta}_i),\tag{1}$$

where $[\hat{n}_i, \hat{\theta}_j] = i\delta_{ij}$, and $\hat{\theta}_j$ prefers to take values $\hat{\theta}_j = 2\pi k/N$ with $k = 0, \cdots N-1$ due to the $u$-term. The potential $V(\hat{n})$ has a minimum at $\hat{n} = 0$. The order-disorder transition of the $Z_N$ symmetry is described by the Landau-Ginzburg action

$$\begin{aligned}
\mathcal{S} &= \int d^2x\, d\tau\; |\partial\Phi|^2 + r|\Phi|^2 + g|\Phi|^4 + u(\Phi^N + h.c.) &\longleftrightarrow \\
\mathcal{S}_d &= \int d^2x\, d\tau\; |(\partial - ia)\phi|^2 + \tilde{r}|\phi|^2 + \tilde{g}|\phi|^4 + u(M^N + h.c.).
\end{aligned}\tag{2}$$

$\Phi$ is the complex order parameter. The second line of the equation is the well-known boson-vortex dual description of the phase transition [36–38], and $r \sim -\tilde{r}$ is the tuning parameter of the transition: $r0$ ($r < 0$) corresponds to the gapped (condensed) phase of $\Phi$ and condensed (gapped) phase of $\phi$. The $\Phi^N$ term is the $Z_N$ anisotropy on $\Phi$ which breaks the U(1) symmetry of $\Phi$ to $Z_N$. The $\Phi^N$ is dual to the $N-$fold monopole operator ($M^N$) in the dual theory. It is known that when $N \geq 4$, the $u$ term ($Z_N$ anisotropy) is an irrelevant perturbation at the $(2+1)d$ XY transition, and there will be an emergent U(1) symmetry at the quantum phase transition.

As was discussed before, a system with $Z_N$ symmetry has an inexplicit dual $\tilde{Z}_N$ 1-form symmetry. One can embed this system to the boundary of a $(3+1)d$ $Z_N$ topological order, and the $Z_N$ and $\tilde{Z}_N^{(1)}$ symmetry can both be made explicit (as is defined the appendix), and they together constitute the "categorical symmetry" of the system [15]. In order to describe the behavior of the $\tilde{Z}_N^{(1)}$ symmetry, Ref. [16] introduced the "order diagnosis operator" $\tilde{O}_{\mathcal{C}}$. Represented in terms of lattice operators, the ODO for the dual $Z_N^{(1)}$ symmetry reads

$$\tilde{O}_{\mathcal{C}} = \exp\left(i\frac{2\pi}{N}\sum_{j\in\mathcal{A}}\hat{n}_j\right),\tag{3}$$

where $\partial\mathcal{A} = \mathcal{C}$ is a patch of the $2d$ lattice enclosed by contractible loop $\mathcal{C}$, and the ODO was also called patch operator in Ref. [15]. $\tilde{O}_{\mathcal{C}}$ creates a $Z_N$ domain wall. In the ordered and disordered phase of the $Z_N$ symmetry, the expectation value of $\tilde{O}_{\mathcal{C}}$ decays with an area law and perimeter law respectively.

At the order-disorder phase transition, to extract the universal feature of the ODO $\tilde{O}_{\mathcal{C}}$, we evaluate $\langle(\log\tilde{O}_{\mathcal{C}})^2\rangle$ [2], which in the dual theory reduces to

$$\langle(\log\tilde{O}_{\mathcal{C}})^2\rangle = -\frac{1}{N^2}\int_{\mathcal{C}}dl^\mu\int_{\mathcal{C}'}dl'^\nu\langle a_\mu(\mathbf{x})a_\nu(\mathbf{x}')\rangle.\tag{4}$$

The relation between $a_\mu$ and the original Landau-Ginzburg theory is $J = \frac{i}{2\pi}*da$, where $J$ is the current of the emergent U(1) symmetry at the $Z_N$ order-disorder transition [3]. The correlation

---

[2]log is a multivalued function. Since $\tilde{O}_{\mathcal{C}} = \prod_j \tilde{O}_{j\in\mathcal{A},\partial\mathcal{A}=\mathcal{C}}$, where $\tilde{O}_j = e^{i2\pi\hat{n}_j/N}$, we define $\log\tilde{O}_{\mathcal{C}} = \sum_{j\in\mathcal{A}}\log\tilde{O}_j$, and demand $\text{Arg}[\tilde{O}_j] = \log\tilde{O}_j \in (-\pi,\pi] \sim 2\pi\hat{n}_j/N$, the $V(\hat{n}_i)$ term in the Hamiltonian Eq. 1 restricts $\hat{n}_j$ to largely fluctuate around its minimum $\hat{n}_j \sim 0$.

[3]In this work we restrict our discussions on systems with $Z_N$ or $Z_N^{(1)}$ symmetry on the lattice. The desired ODO of the disordered phase of a system with a $Z_N$ symmetry on a $2d$ lattice, is a loop object, which can also be viewed as the "disordered operator" [17]. The evaluation of the behavior of the loop object is evaluated in an IR field theory with an emergent U(1) symmetry, but when a system does have a U(1) symmetry on the lattice, the disordered phase is driven by the condensation of vortices, rather than a loop object. The physical meaning of ODO with discrete symmetry is most clear when the lattice symmetry is discrete. Generalization of categorical symmetries to continuous symmetry is possible, but we leave this to more careful future study.

of $a_\mu$ is dictated by the correlation of $J$ whose scaling dimension *does not* renormalize at a general conformal field theory. The correlation between currents $J$ is proportional to the universal conductivity at a $(2+1)d$ conformal field theory:

$$\langle J_\mu(0)J_\nu(\mathbf{x})\rangle = \sigma \frac{I_{\mu\nu}(\mathbf{x})}{|\mathbf{x}|^4}, \tag{5}$$

where the matrix $I_{\mu\nu}(\mathbf{x})$ is given by $I_{\mu\nu}(\mathbf{x}) = \delta_{\mu\nu} - 2x_\mu x_\nu/|\mathbf{x}|^2$, and $\sigma$ is $C_J$ in (for example) Ref. [39]. The universal conductivity at a $(2+1)d$ XY transition was predicted in Ref. [40], and it can be computed using various theoretical and numerical methods, and also measured experimentally (see for example Ref [41–49], the universal conductivity in some of the references was computed/measured with strong disorder).

It is straightforward to verify that the gauge field propagator can be written as

$$\langle a_\mu(0)a_\nu(\mathbf{x})\rangle = \sigma\pi^2 \frac{\delta_{\mu\nu} - \zeta I_{\mu\nu}(\mathbf{x})}{|\mathbf{x}|^2}. \tag{6}$$

The parameter $\zeta$ is introduced by a nonlocal gauge fixing term

$$\frac{1}{8\pi^6\sigma}\frac{1}{1-\zeta}\int d^3\mathbf{x}d^3\mathbf{y}\frac{\partial_\mu a^\mu(\mathbf{x})\partial_\nu a^\nu(\mathbf{y})}{|\mathbf{x}-\mathbf{y}|^2}, \tag{7}$$

which contributes to a total derivative $I_{\mu\nu}(\mathbf{x})/|\mathbf{x}|^2 = \frac{1}{2}\partial_\mu\partial_\nu\log|\mathbf{x}|^2$ in the gauge field propagator.

In the explicit calculation of Eq. 4, one should be very careful about how to set the UV cut-off. A hard cut-off on the integration interval $|\mathbf{x}-\mathbf{x}'|$ along $\mathcal{C}$ will spoil the gauge invariance. To guarantee that $\mathcal{C}$ and $\mathcal{C}'$ are both complete loops in the integral (hence gauge invariance is preserved), a good method is to set a small distance between $\mathcal{C}$ and $\mathcal{C}'$ along the temporal direction by distance $\tau = \epsilon > 0$, and this small splitting serves as a small real-space UV cut-off. The integral is then performed along the closed loop $\mathcal{C}$ (and its duplicate $\mathcal{C}'$) in the $x$-$y$ plane. For a smooth loop $\mathcal{C}$ with perimeter $P$, the evaluation of $\langle(\log O_\mathcal{C})^2\rangle$ simply yields a perimeter law, *i.e.* proportional to $P$ with a UV-dependent coefficient. For example, when $\mathcal{C}$ is a circle with radius $R$, the integral in Eq. 4 gives

$$-\langle(\log \tilde{O}_\mathcal{C})^2\rangle = \frac{\sigma\pi^2}{N^2}\left(\frac{2\pi^2 R}{\epsilon} - 2\pi^2 + \frac{3\pi^2\epsilon}{4R}\right) + \mathcal{O}(\epsilon^2). \tag{8}$$

There are two observations. First, the final result is independent of the gauge choice $\zeta$. Second, the large-$R$ scaling is only given by a linear term which depends on the UV cut-off.

However, if the loop $\mathcal{C}$ has sharp corners, the situation is very different, and some universal feature that does not depend on the UV cut-off emerges. Let us first consider $\mathcal{C}$ being a spatial square with four corners $(0,0),(L,0),(L,L),(0,L)$. There are three types of integrals that are involved. The linear contribution is from the correlation along the same edge of $\mathcal{C}$

$$\int_0^L dx\int_0^L dx'\frac{(1+\zeta)(x-x')^2+(1-\zeta)\epsilon^2}{((x-x')^2+\epsilon^2)^2} = \frac{\pi L}{\epsilon} - 2(1+\zeta)\log(L/\epsilon) + \mathcal{O}(1). \tag{9}$$

It is important to notice that there is a $\log(L/\epsilon)$ term, which also shows up in the integral for two neighboring edges that are perpendicular to each other

$$\int_0^L dx\int_0^L dy'\frac{2\zeta xy'}{(x^2+y'^2+\epsilon^2)^2} = \zeta\log(L/\epsilon). \tag{10}$$

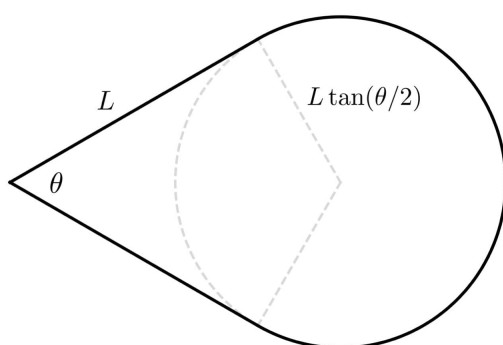

Figure 1: The shape of $\mathcal{C}$ with only one angle $0 < \theta < \pi$. As a concrete example, we consider a circle with two tangent lines that intersect at a point. Each tangent line has the length $L$, the radius of the circle is therefore $L\tan(\theta/2)$ and the perimeter of $\mathcal{C}$ is given by $P = (2 + (\pi + \theta)\tan(\theta/2))L$.

The integral from two parallel edges is a finite number which does not grow with $L$

$$\int_0^L dx \int_0^L dx' \frac{(\zeta+1)(x-x')^2 + (1-\zeta)(L^2+\epsilon^2)}{-(L^2+(x-x')^2+\epsilon^2)^2} = \mathcal{O}(1). \tag{11}$$

Combining all contributions together, we find the gauge invariant result

$$-\langle(\log \tilde{O}_\mathcal{C})^2\rangle = \frac{\sigma \pi^2}{N^2}\left(\frac{\pi 4L}{\epsilon} - 8\log(L/\epsilon)\right) + \mathcal{O}(1). \tag{12}$$

The $\zeta$-independence of the $\mathcal{O}(1)$ term has also been verified. This result is similar to the evaluation of a square Wilson loop for free QED in $(3+1)$ dimensions. In both the two cases above, we find that the linear term in $-\langle(\log \tilde{O}_\mathcal{C})^2\rangle$ is $\frac{\sigma\pi^2}{N^2}\frac{\pi P}{\epsilon}$ where $P = 2\pi R$ for the circle and $P = 4L$ for the square.

Let us now generalize Eq. 10 to the case of two straight lines with an arbitrary angle $\theta$ with $0 < \theta < \pi$. For convenience, we choose the gauge $\zeta = 0$ in the following calculations. We could parametrize the two straight lines by $t(\cos(\theta/2), -\sin(\theta/2))$ and $s(\cos(\theta/2), \sin(\theta/2))$ where $0 < s, t < L$. To extract the angle-dependence of the logarithmic divergence, we use the trick in Ref. [50, 51]

$$\int_0^L ds \int_0^L dt \frac{-\cos\theta}{s^2 + t^2 - 2st\cos\theta + \epsilon^2} =$$
$$\int_0^L d\ell \int_0^1 d\lambda \left[\frac{\ell}{\ell^2 + \epsilon^2}\frac{-\cos\theta}{\lambda^2 + (1-\lambda)^2 - 2\lambda(1-\lambda)\cos\theta} + \mathcal{O}(\epsilon^2/\ell^3)\right],$$

where we have changed the integration variables to $s = \ell\lambda$, $t = \ell(1-\lambda)$, and the $\mathcal{O}(\epsilon^2/\ell^3)$ part does not contribute to any logarithmic divergence. The $\lambda$-integral can be evaluated exactly, which gives $-(\pi - \theta)\cot\theta$. The $\log(L/\epsilon)$ divergence then arises from the $\ell$-integral. There is another logarithmic contribution from correlation within the same line. Combining all the contributions together, eventually we obtain

$$-\langle(\log \tilde{O}_\mathcal{C})^2\rangle = \frac{\sigma\pi^2}{N^2}\left(\frac{\pi P}{\epsilon} - f(\theta)\log P\right) + \mathcal{O}(1), \tag{13}$$
$$f(\theta) = 2(1 + (\pi - \theta)\cot(\theta)), \tag{14}$$

for any shape of $\mathcal{C}$ with a single corner, where $P$ is the perimeter of $\mathcal{C}$. We observe that the universal logarithmic term vanishes when $\theta = \pi$, and only the linear term remains, as expected.

| $\theta$ | $L$ | $\log L$ | $1/L$ | const. |
|---|---|---|---|---|
| $\pi/20$ | 7.10018 | -40.0651 | 14.6678 | 38.1625 |
| $\pi/10$ | 8.00282 | -19.4447 | 21.0523 | -1.35483 |
| $3\pi/20$ | 9.00801 | -12.474 | 17.0716 | -10.7626 |
| $\pi/5$ | 10.1315 | -8.93462 | 12.4601 | -14.3023 |
| $\pi/4$ | 11.3933 | -6.70427 | 11.2763 | -16.2376 |

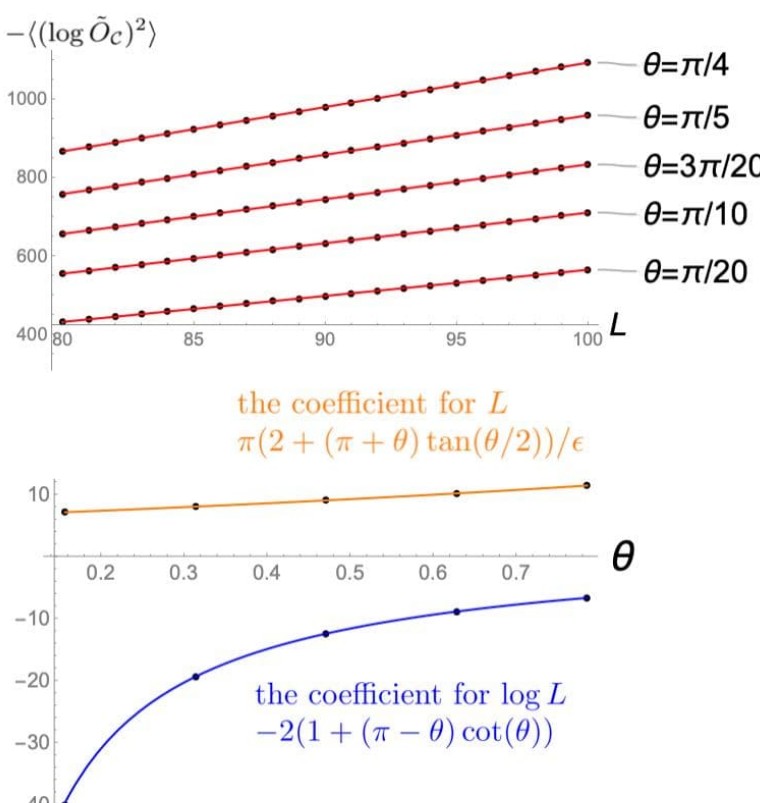

Figure 2: The numerical results of $-\langle(\log \tilde{O}_{\mathcal{C}})^2\rangle$ (in the unit of $\sigma\pi^2/N^2$) for the shape in FIG. 1 with different angles. The UV cut-off is set to be $\epsilon = 1$. The large-$L$ scaling is fitted by the function $-\langle(\log \tilde{O}_{\mathcal{C}})^2\rangle = aL/\epsilon + b\log L + c/L + d$, and the fitting parameters $a, b$ agree with the analytical expressions Eq. 13 and Eq. 14.

To double check the analytical expression Eq. 13, we consider the shape of $\mathcal{C}$ as shown in FIG. 1, and the numerical evaluation for $-\langle(\log \tilde{O}_{\mathcal{C}})^2\rangle$ for different angles are shown in FIG. 2. For fixed values of $L, \epsilon$, the angle dependence for both the linear and the logarithmic terms agree with Eq. 13 and Eq. 14.

We computed $-\langle(\log \tilde{O}_{\mathcal{C}})^2\rangle$, which is the second order expansion of $2\langle \tilde{O}_{\mathcal{C}}\rangle$. We have not proven whether higher order expansion in $\langle \tilde{O}_{\mathcal{C}}\rangle$ leads to different corner contribution from $\langle(\log \tilde{O}_{\mathcal{C}})^2\rangle$ or not. We would also like to mention that the entanglement entropy of a patch $\mathcal{A}$ with corners in a $(2+1)d$ CFT is related to another universal quantity $C_T$ from the correlation of the stress-energy tensor $T_{\mu\nu}$. As discussed in Ref. [20–24], the entanglement entropy takes the form $S = \frac{B}{\epsilon}P - a(\theta)\log P + \mathcal{O}(1)$, where $B/\epsilon$ depends on the UV details, and the universal coefficient $a(\theta)$ is given by the correlations of $T_{\mu\nu}$.[4] The function $a(\theta)$ proposed and computed for entanglement entropy [20, 21] is also proportional to $f(\theta)$ in our result.

---

[4]The leading order contribution to $a(\theta)$ is given by $C_T$; contribution from higher order correlations between $T_{\mu\nu}$ was discussed in Ref. [24].

## 2.2 Example 2: $Z_N$ SPT-trivial transition

Now let us still assume the system has a $Z_N$ symmetry, but the system undergoes a transition between a 2$d$ $Z_N$ symmetry protected topological (SPT) state and a trivial state. Both states are disordered states of the $Z_N$ symmetry, hence in both states the ODO $\tilde{O}_C$ should obey a perimeter law. Our main interest focuses on the trivial-SPT phase transition, especially the universal features of $\tilde{O}_C$ at this transition. This example, and the next few examples will be described by a class of similar theories:

$$\mathcal{S} = \int d^2x d\tau \, \sum_{\alpha=1}^{N_f} \bar{\psi}_\alpha \gamma \cdot (\partial - ina)\psi_\alpha + m\bar{\psi}\psi + \frac{ik}{4\pi}ada + \cdots, \tag{15}$$

with integer $N_f$ and $n$, and in general these theories will be labelled as $\text{QED}_{(N_f,n,k)}$. The trivial-SPT transition corresponds to $\text{QED}_{(2,1,0)}$, i.e. $N_f = 2$, $n = 1$ and $k = 0$ [52, 53], plus Chern-Simons terms of background gauge fields which are not written explicitly in Eq. 15. The trivial-SPT transition needs certain fine-tuning to reach the critical point described by this field theory, hence this field theory is a multi-critical point between the two states. This multi-critical point is self-dual [54–56] and also dual to the easy-plane deconfined quantum critical point [13, 14, 57, 58]. The Dirac fermion mass term $m$ in Eq. 15 is the tuning parameter between the trivial and SPT phases.

In the theory $\text{QED}_{(2,1,0)}$, the current of the U(1) symmetry in which the microscopic $Z_N$ symmetry is embedded, is $J = \frac{i}{2\pi} * da$, and the ODO of the system is given by Eq. 3. The angle dependence of the ODO is still give by Eq. 14, with $\sigma$ replaced by the counterpart at the trivial-SPT (multi-)critical point $\text{QED}_{(2,1,0)}$. The universal conductivity can be computed using various methods such as $1/N_f$ expansion.

# 3 Systems with explicit $Z_N^{(1)}$ symmetry

## 3.1 Topological transition at the boundary of a 3$d$ SPT with $Z_N^{(1)} \times \text{U}(1)^{(0)}$ symmetry

Here we consider an example with an explicit $Z_N^{(1)}$ 1-form symmetry. The infrared of this example is described by $\text{QED}_{(1,2N,0)}$ of Eq. 15, i.e. it is a single massless Dirac fermion $\psi$ with charge$-2N$ coupled with a U(1) gauge field. In our construction of theory $\text{QED}_{(1,2N,0)}$ we also need a charge$-N$ fermion $\psi'$ in the background, hence the system only has a $Z_N^{(1)}$ 1-form symmetry, i.e. the electric flux of the gauge field through any closed surface is conserved mod $Z_N$. We also demand that the magnetic flux of the $\text{QED}_{(1,2N,0)}$ is conserved, which corresponds to another U(1)$^{(0)}$ symmetry. There is a mixed anomaly between the $Z_N^{(1)}$ and U(1)$^{(0)}$ symmetries. Hence the field theory $\text{QED}_{(1,2N,0)}$ can be realized at the boundary of a 3$d$ SPT state with $Z_N^{(1)}$ and U(1)$^{(0)}$ symmetry [59]. In the following paragraphs we spell out this construction of the 3$d$ bulk SPT state. [5]

To construct the boundary theory $\text{QED}_{(1,2N,0)}$, we first consider a 3$d$ bulk with an ordinary photon phase of gauge field $a_\mu$, and only charge$-N$ and charge$-2N$ fermionic matter field is dynamical, although all the integer-charge Wilson loops are allowed in the theory. Hence the system has a $Z_N^{(1)}$ 1-form symmetry. All the fermionic matters are in a topologically trivial band structure in 3$d$. Then we bind the Dirac monopole of $\vec{a}$ with another gauge neutral boson with global U(1)$^{(0)}$ conservation, and condense the bound state. The 3$d$ bulk is a SPT state with

---

[5]This is one possible construction of the 3$d$ bulk, the field theory $\text{QED}_{(1,2N,0)}$ maybe realized as the boundary theory of other 3$d$ 1-form SPT states too.

$Z_N^{(1)} \times U(1)^{(0)}$ symmetry [59]. The natural $2d$ boundary of the system is a $(2+1)d$ photon phase. To create a gauge flux at the $2d$ boundary, one needs to move a Dirac monopole from outside of the system, into the $3d$ bulk; since in the $3d$ bulk the bound state between the Dirac monopole and the $U(1)^{(0)}$ boson is condensed, the $2\pi$ magnetic flux at the boundary must also carry the $U(1)^{(0)}$ boson. Hence the photons at the $2d$ boundary is the dual of the Goldstone modes of the $U(1)^{(0)}$ symmetry. Notice that the bulk is fully gapped and has no spontaneous breaking of the $U(1)^{(0)}$ symmetry, because the condensed bound state in the bulk is coupled to the dual gauge field while carrying the $U(1)^{(0)}$ charge. The condensate is still gapped due to the Higgs mechanism.

At the $2d$ boundary, the charge$-2N$ fermion $\psi$ is tuned close to the transition between a trivial insulator and a Chern insulator with Chern number $+1$. Due to the fermi-doubling in $2d$, there must be another massive Dirac cone of $\psi$ in the band structure that affects the dynamics of $a_\mu$. Hence we need to design a background band structure of the charge$-N$ fermion $\psi'$ with Chern number $-2$. The Chern-Simons term of $a_\mu$ generated from $\psi'$ will cancel the Chern-Simons term generated by the band structure of fermion $\psi$.

Now we have arrived at the theory $QED_{(1,2N,0)}$. The $QED_{(1,2N,0)}$ is a transition between two different topological states tuned by the mass of the Dirac fermion $\psi$, these two topological orders are described by the CS term for $a_\mu$ with level $k = \pm 2N^2$, which is free of $Z_N^{(1)}$ 1-form symmetry anomaly. The ODO for the $Z_N^{(1)}$ symmetry is the charge-1 Wilson loop $O_{\mathcal{C}} = \exp(i \int d\vec{l} \cdot \vec{a})$. In this case the quantity $\langle (\log O_{\mathcal{C}})^2 \rangle$ at the critical point $m = 0$ can be evaluated exactly, based on the fermion-vortex duality developed recently [60–64]:

$$QED_{(1,2N,0)} \;\longleftrightarrow\; \bar{\chi}\gamma \cdot \partial \chi \text{ coupled to } Z_N \text{ gauge theory} + \cdots \tag{16}$$

The detailed and exact form of the duality can be found in Ref. [64]. The right hand side of the duality is a Dirac fermion coupled with a $Z_N$ gauge field. The duality relation we will exploit is

$$J_\chi = i \frac{2N}{4\pi} * da, \tag{17}$$

where $J_\chi$ is the current carried by $\chi$. Although $\chi$ is coupled with a $Z_N$ gauge field, since the $Z_N$ gauge field is gapped, in the infrared the correlation of $J_\chi$ is identical to that of the free Dirac fermion, and can be computed exactly:

$$\left\langle J_{\chi,\mu}(0) J_{\chi,\nu}(\mathbf{x}) \right\rangle = \frac{1}{8\pi^2} \frac{I_{\mu\nu}(\mathbf{x})}{|\mathbf{x}|^4}. \tag{18}$$

One can determine the propagator of the dual gauge field accordingly. Considering again the $\mathcal{C}$ in FIG. 1, we find

$$-\langle (\log O_{\mathcal{C}})^2 \rangle = \frac{1}{8N^2}\left( \frac{\pi P}{\epsilon} - f(\theta) \log P \right) + \mathcal{O}(1), \tag{19}$$

where $f(\theta)$ is given in Eq. 14.

## 3.2 $QED_{(N_f,N,k)}$ with explicit $Z_N^{(1)}$ symmetry and Chern-Simons term

We consider the theory $QED_{N_f,N,k}$ with large$-N_f$ and level $k = qN^2$, where $q$ is an integer at the order of $N_f$. $QED_{(N_f,N,k)}$ with even integer $N_f$, and a CS term with level $k$ being integer multiple of $N^2$ can be constructed in $2d$ with $Z_N^{(1)}$ 1-form symmetry [6]. At low energy, the dynamics of gauge field is significantly modified by the one-loop polarization diagram of fermion

---

[6]We can verify that the absence of the anomaly associated to the $Z_N$ 1-form symmetry in this QED theory by considering the its massive phases. For example, when a positive mass of the Dirac fermion is turned on, one

$\psi$. In the momentum space, the loop diagram integral gives

$$|a_\mu(\vec{p})|^2 \frac{N_f N^2}{16} \frac{|p|^2 \delta_{\mu\nu} - p_\mu p_\nu}{|p|}, \tag{20}$$

which gives an order $N_f$ contribution to the gauge field self-energy. To the leading order in $1/N_f$, the gauge field propagator in the momentum space is given by

$$\frac{16}{N_f N^2} \frac{1}{|p|} \left( \frac{\cos \hat{K}}{|K|} \left( \delta_{\mu\nu} - \zeta \frac{p_\mu p_\nu}{|p|^2} \right) + \frac{\sin \hat{K}}{|K|} \frac{\varepsilon_{\mu\nu\sigma} p_\sigma}{|p|} \right), \tag{21}$$

where $|K|, \hat{K}$ denote the magnitude and the angle of the two-dimensional vector $K = (1, \frac{-16k}{2\pi N_f N^2})$. The Fourier transformation to real space gives

$$\langle a_\mu(0) a_\nu(\mathbf{x}) \rangle = \frac{8}{N_f N^2} \frac{1}{\pi^2 |\mathbf{x}|^2} \times \left( \frac{\cos \hat{K}}{|K|} \frac{\delta_{\mu\nu} - \zeta I_{\mu\nu}(\mathbf{x})}{|\mathbf{x}|^2} + \frac{\sin \hat{K}}{|K|} \frac{i\pi}{2} \frac{\varepsilon_{\mu\nu\sigma} x_\sigma}{|\mathbf{x}|} \right),$$

which has an imaginary part due to the Chern-Simons term. The parameter $\zeta$ is introduced by gauge fixing.

The ODO for the $Z_N^{(1)}$ symmetry is still the charge-1 Wilson loop $O_\mathcal{C} = \exp(i \int d\vec{l} \cdot \vec{a})$. As for the shape of $\mathcal{C}$ with a sharp corner in FIG. 1, our calculation leads to the gauge invariant result

$$-\langle (\log O_\mathcal{C})^2 \rangle = \frac{8N^2 N_f}{64k^2 + \pi^2 N^4 N_f^2} \left( \frac{\pi P}{\epsilon} - f(\theta) \log P \right) + \mathcal{O}(1), \tag{22}$$

where $f(\theta)$ is given in Eq. 14. The imaginary antisymmetric part of $\langle a_\mu a_\nu \rangle$ does not contribute, and the final result has the similar form as before. In the large$-N_f$ limit the universal conductivity of the current $J = \frac{1}{2\pi} * da$ can be computed exactly.

# 4 The "Strange Correlator" of ODO

Following the argument from Ref. [65], if a state $|\Omega\rangle$ is the ground state described by a Lagrangian $\mathcal{L}(\Phi(\mathbf{x}))$, the matrix elements between $|\Omega\rangle$ and two different field configurations $|\Phi(\mathbf{x})\rangle$ and $|\Phi'(\mathbf{x})\rangle$ is given by the path integral:

$$\langle \Phi(\mathbf{x}) | \Omega \rangle \langle \Omega | \Phi'(\mathbf{x}) \rangle \sim \int_{\Phi(\mathbf{x}, \tau=-\infty)=\Phi'(\mathbf{x})}^{\Phi(\mathbf{x}, \tau=+\infty)=\Phi(\mathbf{x})} D\Phi(\mathbf{x}, \tau) \times \exp \left( -\int_{-\infty}^{+\infty} d\tau d^d x \, \mathcal{L}(\Phi(\mathbf{x}, \tau)) \right), \tag{23}$$

knowing the matrix element, Ref. [65] was able to derive the ground state wave function based on the Lagrangian description of various SPT states.

Based on the information of the ground state wave function of SPT state derived from its Lagrangian, the quantity "strange correlator" was introduced and designed to diagnose

---

obtains a U(1) CS theory of level $(q + N_f/2)N^2$. In this massive phase, the $Z_N$ 1-form symmetry is generated by the anyon line operator carrying U(1) charge $(q + Nf/2)N$. When $N$ is odd, we should in fact view the U(1) gauge field $a$ as a spin$_c$ gauge field. Consequently, this charge-$(q+N_f/2)N$ anyon always has bosonic self-statistics, which indicates the absence of anomaly associated with the $Z_N$ 1-form symmetry. When $N$ is even, the QED (and its massive phases) intrinsically resides in a fermionic Hilbert space. The gauge field $a$ is now a regular U(1) gauge field. In this case, the charge-$(q + N_f/2)N$ anyon can have either bosonic or fermionic self-statistics depending on the value of $(q + N_f/2)N$. However, neither case leads to any anomaly associated to the $Z_N$ 1-form symmetry because the self-statistics of the charge-$(q+N_f/2)N$ anyon can be made bosonic by attaching extra neutral fermions in the Hilbert space.

a SPT state based on its bulk wave function [26]. Let us assume that $|0\rangle$ and $|1\rangle$ are the trivial state and SPT state defined within the same bosonic Hilbert space in a two dimensional real space, and both systems have the same symmetry. The strange correlator is the quantity $S(\mathbf{x}, \mathbf{x}') = \langle 0|\Phi(\mathbf{x})\Phi(\mathbf{x}')|1\rangle/\langle 0|1\rangle$, where $\Phi(\mathbf{x})$ is the order parameter of the symmetry that defines the systems.

For a class of Langrangians $\mathcal{L}$, using the derived wave functions for both the SPT state $|1\rangle$ and trivial state $|0\rangle$, one would see that the strange correlator $S(\mathbf{x}, \mathbf{x}')$ cannot have a trivial short range correlation at least for $d = 2$. Another picture to see this is that, if the Lagrangian $\mathcal{L}$ has an emergent Lorentz invariant description, after the space-time rotation, the strange correlator which was purely defined in space, becomes a space-time correlation function at the one dimensional spatial interface between $|0\rangle$ and $|1\rangle$. This picture is similar to the construction of fractional quantum Hall wave function using conformal blocks [66]. Because the spatial interface between $|0\rangle$ and $|1\rangle$ cannot be trivially gapped, the strange correlator $S(\mathbf{x}, \mathbf{x}')$ must be either long ranged, or have a power-law. Hence the strange correlator can be viewed as a tool to diagnose a SPT state based on its bulk wave function, and it has been shown to be effective for many examples [27–35].

ODO is the generalization of correlation functions of 0-form symmetries. Here we generalize the strange correlator to the ODO of 1-form symmetry i.e. we evaluate the following quantity

$$S(\mathcal{C}) = \langle 0|O_{\mathcal{C}}|1\rangle/\langle 0|1\rangle, \tag{24}$$

where $|0\rangle$ and $|1\rangle$ are trivial state and SPT state with 1-form symmetry respectively. SPT states protected by 1-form symmetries have attracted great interests in the last few years [7, 9, 59, 67–76], we expect this general question of evaluating strange correlator of ODO to be a new direction that is worth a deep exploration. In the current work we consider a typical $3d$ SPT state protected by the $Z_N^{(1)}$ 1-form symmetry as an example. This SPT state can be described by the following Lagrangian [77]

$$\mathcal{L} = \frac{1}{g}\text{tr}[F_{\mu\nu}F_{\mu\nu}] + \frac{i\Theta}{8\pi^2}\text{tr}[F \wedge F]. \tag{25}$$

$F$ is the curvature tensor of the SU($N$) gauge field. To guarantee there is a $Z_N^{(1)}$ 1-form symmetry, we only allow dynamical (but massive) matter fields of the SU($N$) gauge field which carries an adjoint representation of the gauge field, while closed Wilson loops with other representations of the gauge field are still allowed. The SPT state corresponds to $\Theta = 2\pi$, while the trivial state corresponds to $\Theta = 0$ in the Lagrangian. The interface between $\Theta = 0$ and $\Theta = 2\pi$ is a $2d$ topological order described by SU($N$)$_1$ Chern-Simons theory with topological degeneracy. For both $\Theta = 0$ or $2\pi$, the coupling constant $g$ in the Lagrangian is expected to flow to infinity under renormalization group, hence the $\Theta-$term is what remains in the infrared limit. The $\Theta-$term is a total derivative, hence

$$
\begin{aligned}
\langle A(\mathbf{x})|1\rangle\langle 1|A'(\mathbf{x})\rangle \quad &\sim \quad \int_{A(\mathbf{x},\tau=-\infty)=A'(\mathbf{x})}^{A(\mathbf{x},\tau=+\infty)=A(\mathbf{x})} DA(\mathbf{x},\tau) \times \exp\left(-\int_{-\infty}^{+\infty} d\tau d^3x \, \mathcal{L}(A)_{g\to+\infty}\right) \\
&\sim \quad \exp\left(\int d^3x \frac{i}{4\pi}\text{CS}[A] - \frac{i}{4\pi}\text{CS}[A']\right).
\end{aligned}
\tag{26}
$$

Hence the wave function of the SPT state $|1\rangle$, and the trivial state $|0\rangle$ (corresponds to $\Theta = 0$) in the limit $g \to +\infty$ are schematically

$$|0\rangle \quad \sim \quad \int DA|A\rangle,$$

$$|1\rangle \quad \sim \quad \int DA \exp\left(\int d^3x \frac{\mathrm{i}}{4\pi} \mathrm{CS}[A]\right)|A\rangle. \tag{27}$$

Now the evaluation of the strange correlator of ODO, which is a purely $3d$ spatial quantity, is mathematically equivalent to evaluating world lines of anyons in $(2+1)d$ $\mathrm{SU}(N)_1$ CS field theory:

$$S(\mathcal{C}) \sim \int DA \, \mathrm{tr}[e^{\mathrm{i}\int_{\mathcal{C}} d\vec{l}\cdot\vec{A}}] \exp\left(\int d^3x \frac{\mathrm{i}}{4\pi} \mathrm{CS}[A]\right). \tag{28}$$

Then if the ODO is a Wilson loop with the fundamental representation of the gauge group, and $\mathcal{C}$ contains two loops with a link, then this evaluation is identical to the braiding process of two anyons of the $\mathrm{SU}(N)_1$ topological order, and it yields phase $\exp(\mathrm{i}2\pi/N^2)$ for $S(\mathcal{C})$.

## 5 Discussion

In this work we studied the behavior of the "order diagnosis operator" of 1-form symmetries (for either explicit 1-form symmetry, or inexplicit 1-form symmetry as a dual of a 0-form symmetry) at various $(2+1)d$ quantum phase transitions. We demonstrate that for a class of transitions there is a universal logarithmic contribution to the ODO arising from the corners of the loop upon which the ODO is defined. For this class of transitions, the universal logarithmic contribution is related to the universal conductivity at the critical points, and in some cases can be computed exactly using the duality between conformal field theories.

This logarithmic contribution is similar to the corner contribution to the entanglement entropy, in fact this relation can be made exact for free boson/fermion systems [19]. For general systems, the ODO associated with certain 1-form symmetry and the entanglement entropy can be studied in a unified framework. To study the Renyi entropy, one needs to use the replica trick, and duplicate $n-$copies of the system. Then the system is granted an extra "swapping symmetry" between replica indices. The Renyi entropy reduces to evaluating the ODO of the 1-form dual of the swapping symmetry [78, 79]. Hence we can start with the duplicated system, and just study the ODO of all the symmetries of the duplicated system, to extract the information of both the intrinsic symmetries, and the entanglement entropy simultaneously. One remark worth making is that, when computing Renyi entropy for ordinary systems with a Hamiltonian and translation invariance, there is no interaction between different duplicated systems, hence each duplicated copy has its own conservation laws. [7]

In this work we also computed the strange correlator of the 1-form ODO for a particular example. SPT states protected by 1-form symmetries have attracted great efforts and interests in the last few years, and we believe the strange correlator of the 1-form ODO can be applied to many related systems. We will leave the more general discussion of this topic to future studies.

The authors thank Wenjie Ji and Yi-Zhuang You for very helpful discussions. This work is supported by NSF Grant No. DMR-1920434, the David and Lucile Packard Foundation, and the Simons Foundation.

*Note:* We would like to draw the readers attention to a closely related work by Yan-Cheng Wang, Meng Cheng and Zi Yang Meng [81] to appear in the same arXiv listing.

---

[7]The authors note that a more recent work Ref. [80] demonstrated the corner contribution for correlation functions integrated over an area is very universal, which bridged the ODO considered here and the entanglement entropy on general grounds.

# A   Clarification of Concepts

The purpose of this appendix is not to discuss new physics or new quantity, but to clarify the rudimentary concepts used in this manuscript. The standard definition of a global symmetry of a quantum system is associated with a global conserved quantity $\hat{G}$ that commutes with the entire Hamiltonian of the system. Normally when we say a system has a global symmetry, it implies the following two qualities of the system:

*(1)* the dynamics allowed by the symmetry, for example the evolution generated by the Hamiltonian of the system does not change the quantum number of quantity $\hat{G}$;

*(2)* states with different quantum numbers of $\hat{G}$ are all present in the Hilbert space.

To exemplify these two qualities, let us still start with the basic example of $1d$ quantum Ising model with a transverse field: $H = \sum_j -K\sigma_j^z \sigma_{j+1}^z - h\sigma_j^x$. Here the conserved quantity of the $Z_2$ Ising spin symmetry is $\hat{G} = \prod_j \sigma_j^x$, and any physical process allowed by the symmetry does not change the quantum number of $\hat{G}$ (only processes that flip even number of spins $\sigma_j^x$ are allowed); but states with $\hat{G} = \pm 1$ all exist in the Hilbert space. Hence both qualities *(1)* and *(2)* mentioned above are perfectly satisfied by the $Z_2$ spin symmetry.

It is often stated that the $1d$ quantum Ising model is "self-dual" under the Kramers-Wannier duality, namely if we introduce dual operators $\tau_{\bar{j}}^{z,x}$ as $\sigma_j^z \sigma_{j+1}^z = \tau_{\bar{j}}^x$, $\sigma_j^x = \tau_{\bar{j}-1}^z \tau_{\bar{j}}^z$, the Hamiltonian of the dual model formally takes the form $H = \sum_{\bar{j}} -K\tau_{\bar{j}}^x - h\tau_{\bar{j}}^z \tau_{\bar{j}+1}^z$. Physically $\tau^x$ is the kink of the original operator $\sigma^z$. There appears to be another dual $\tilde{Z}_2$ symmetry, whose conserved quantity $\hat{\tilde{G}}$ is formally $\prod_{\bar{j}} \tau_{\bar{j}}^x$. However, if we take a periodic boundary condition of the original quantum Ising model, $\hat{\tilde{G}}$ is a trivial quantity in the original Ising spin Hilbert space, because $\hat{\tilde{G}}$ always equals to $+1$, or in other words within the original Ising spin Hilbert space, only states with even number of kinks are allowed. Hence although the "$\tilde{Z}_2$ symmetry" satisfies quality *(1)* above, it does NOT meet quality *(2)*.

The dual "$\tilde{Z}_2$ symmetry", though does not meet quality (2), still leads to nontrivial conservation law of kinks of $\sigma^z$: the kink number is unchanged under any physical process for the Ising model with periodic boundary condition. As was pointed out by previous references such as Ref. [15], both the $Z_2$ and $\tilde{Z}_2$ can be made real symmetries (meaning they both satisfy qualities *(1)* and *(2)*) if we embed the $1d$ quantum Ising model as the boundary of a $2d$ toric code model (of course, there were other previously known ways such as introducing different boundary conditions to interpret the $\tilde{Z}_2$ symmetry, but introducing the bulk as Ref. [15] has the most natural generalizations to higher dimensions and higher form dimensions). The Ising spin excitation corresponds to the $e$ anyon of the toric code, and the kink corresponds to the $m$ anyon. The two sets of conservation laws (quality *(1)*) of the Ising spins and kinks arise from the fusion rules of the anyons: $e \times e = I$, $m \times m = I$; now both the $Z_2$ and $\tilde{Z}_2$ symmetries also satisfy quality *(2)*: both the Ising spin number and the kink number can be either even or odd at the $1d$ boundary, because one can create a pair of $e$ (or $m$) anyons, and move only one anyon of the pair to the $1d$ boundary.

Since the original quantum Ising model has conservation laws for dynamics of both the Ising spins and the kinks, in our main text we call the original $Z_2$ spin symmetry of the quantum Ising model as an explicit symmetry (meaning quality *(1)* and *(2)* are both satisfied), while the $\tilde{Z}_2$ symmetry is called an "inexplicit symmetry", as only quality *(1)* is satisfied. As we mentioned in the last paragraph, both $Z_2$ and $\tilde{Z}_2$ symmetries can be made explicit by embedding the system to the boundary of a $2d$ toric code model.

These definitions and notions can be generalized to higher dimensions with higher form discrete symmetries. As a practice let us also consider the $2d$ quantum $Z_2$ gauge theory, which is often stated to be dual to a $2d$ quantum Ising model, though these two models have different

symmetries. To clarify what this duality means exactly, we consider the standard Hamiltonian for the $2d$ quantum $Z_2$ gauge theory on a $2d$ torus: $H = \sum_\square -K \prod_{<ij> \in \square} \sigma_{ij}^z - \sum_{<ij>} h\sigma_{ij}^x$, where $<ij>$ is a link of a square lattice; $\sigma_{ij}^{z,x}$ is a qubit defined on the link. $\prod_{<ij> \in \square} \sigma_{ij}^z$ is a product of $\sigma_{ij}^z$ on the four links around each square plaquette. The Hilbert space of the quantum $Z_2$ gauge theory is subject to a local constraint $\prod_{<ij> \in v} \sigma_{ij}^x = +1$, where $<ij> \in v$ represent four links around a vertex/site of the square lattice. This model has a $Z_2^{(1)}$ 1-form symmetry, which corresponds to the $Z_2$ conservation of $Z_2$ electric field penetrating any contractible loop $\mathcal{C}$: $\hat{G}_{\mathcal{C}} = \prod_{<ij> \perp \mathcal{C}} \sigma_{ij}^x$ ($<ij> \perp \mathcal{C}$ corresponds to all the links on loop $\mathcal{C}$ and orthogonal to $\mathcal{C}$ locally). But if the system is a torus, then $\hat{G}_{\mathcal{C}}$ for a noncontractible loop $\mathcal{C}$ can take values $\pm 1$, which can be interpreted as either the topological sector, or the ground state degeneracy of spontaneous breaking of the $Z_2^{(1)}$ 1-form symmetry. Hence the $Z_2^{(1)}$ 1-form symmetry is an explicit symmetry that satisfies both *(1)* and *(2)* mentioned previously.

The dual $2d$ quantum Ising model can be formally derived by introducing the dual operators on the dual lattice sites $\bar{i}$ and $\bar{j}$, which are located on the center of the plaquette squares of the original square lattice: $\tau_{\bar{i}}^x = \prod_{<ij> \text{around } \bar{i}} \sigma_{ij}^z$, $\tau_{\bar{i}}^z \tau_{\bar{j}}^z = \sigma_{ij}^x$ for $<\bar{i}\bar{j}> \perp <ij>$. The dual Hamiltonian reads $H = \sum_{\bar{i}} -K\tau_{\bar{i}}^x - \sum_{<\bar{i}\bar{j}>} h\tau_{\bar{i}}^z \tau_{\bar{j}}^z$. However, the conserved quantity of the dual Ising model $\tilde{\hat{G}} = \prod_{\bar{i}} \tau_{\bar{i}}^x$ is always $+1$ in the original Hilbert space of the $Z_2$ gauge theory, although a physical process can only create even number of $\tau_{\bar{i}}^x$ (which corresponds to the $m$ anyon of the original quantum $Z_2$ gauge theory) hence there is a $Z_2$ conservation of $\tau^x$. Therefore the dual Ising model has a $\tilde{Z}_2$ symmetry that satisfies quality *(1)* but not *(2)*, hence according to our convention it is an inexplicit symmetry.

Let us also discuss the converse example, and start with a real $2d$ quantum Ising spin model on a square lattice: $H = \sum_{<i,j>} -K\sigma_i^z \sigma_j^z - \sum_j h\sigma_j^x$, which is formally dual to a $2d$ quantum $Z_2$ gauge theory, with the electric field defined on the dual link $<\bar{i}\bar{j}> \perp <ij>$ as $\tau_{\bar{i}\bar{j}}^x = \sigma_i^z \sigma_j^z$. The $2d$ quantum Ising model also has two sets of conservation laws: the conservation of the original Ising spin, and the conservation law of the Ising domain walls. The latter corresponds to a $\tilde{Z}_2^{(1)}$ 1-form "inexplicit symmetry": there is a conservation law of the dynamics of Ising domain wall, namely the Ising domain walls always penetrate any closed contractible loop even times (quality *(1)*); but within the Ising spin Hilbert space the product of $\tau_{\bar{i}\bar{j}}^x = \sigma_i^z \sigma_j^z$ with $<ij> \perp <\bar{i}\bar{j}>$ is always $+1$ along a noncontractible cycle $\mathcal{C}$ orthogonal to the dual lattice link $<\bar{i}\bar{j}>$. But for a real $2d$ $Z_2$ gauge theory, as we discussed above, the corresponding product of electric field can take value $\pm 1$, which can be either interpreted as different topological sectors, or as ground state degeneracy caused by spontaneous breaking of the $Z_2^{(1)}$ 1-form symmetry. Hence in the Ising spin Hilbert space, only the $Z_2$ symmetry satisfies qualities *(1)* and *(2)* together, while $\tilde{Z}_2^{(1)}$ satisfies *(1)* only. But both $Z_2$ and $\tilde{Z}_2^{(1)}$ can be made explicit symmetries, i.e. they can satisfy both *(1)* and *(2)* when the quantum Ising model is embedded as the boundary of a $3d$ topological order.

The quantity order diagnosis operator (ODO) was introduced in Ref. [82] to characterize the behavior of the explicit and inexplicit symmetries, especially the notion of spontaneous symmetry breaking of both the explicit and the inexplicit symmetries defined above. The ODO reduces to previously introduced concepts in specific cases. For example, for the Ising models, the ODO of the dual inexplicit symmetry is the disorder operator discussed in Ref. [17]. But the phrase "disorder operator" implies that when it condenses, the original symmetry would be restored or the system should enter a disordered phase of the original symmetry. This is indeed true for the Ising spin models. But in some cases that involve higher form symmetries both the symmetry and the dual symmetry can be spontaneously broken simultaneously, namely both the explicit symmetry and its dual inexplicit symmetry can enter the ordered phase simultaneously under proper generalizations. For example, a $(3+1)d$ system with $Z_2^{(1)}$ 1-form

symmetry can enter a gapless photon phase where the Wilson loop and the corresponding "disorder operator" of the $Z_2^{(1)}$ 1-form symmetry both have perimeter laws, which is the criterion of spontaneous symmetry breaking of 1-form symmetries. Hence we feel a generalized notion is necessary. In fact, a notion of "patch operator" was introduced in Ref. [15] as a generalization of the the disorder operator to higher form symmetries. The notion of order diagnosis operator used in this manuscript also reduces to the "patch operator" in Ref. [15] for systems without subsystem symmetries. But for systems with a more exotic subsystem symmetries [82] the proper form of the ODO is not always defined on a simple patch of the lattice.

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
