# Peer review of "Universal Features of Higher-Form Symmetries at Phase Transitions"

_SciPost Physics, doi:SciPost Phys. 11, 033 (2021)_

## Round 1 · Referee Report · Anonymous · 2021-3-23

Strengths

1. interesting and timely subject
2. clear presentation of results

Weaknesses

1. some discussion is a bit terse and could be expanded

Report

This is overall a good paper, and I think it is fit to be published in this journal.

I have a few questions:

1. In the introduction (and also in these related works Refs 19 and 78) the authors discuss the behavior of <O_C>. Is there a simple relation between this operator and the operator <(log(O_C))^2>, which is studied in the paper?

2. What numerical methods were used to obtain figure 2? In Ref 19 the authors studied a square loop on a square lattice (using QMC), but I don't see how to get the shape depicted in figure 1.

3. I am a little puzzled by the notion of "categorical symmetry". Suppose I have a system with a global Zn symmetry. I guess you would like to say that the symmetries of the Zn orbifold should be considered as "symmetries" of the original theory? But operators with Zn charge in the original theory are projected out when we go to the orbifold, so they don't have well-defined charges under these other "symmetries". Is it clear this is a well-defined concept?

Some minor comments:

1. It seems to me that what is called an "ODO" operator here is known as a disorder operator since Kadanoff.

2. The divergent 1/epsilon dependent piece of eqn (8) can be eliminated by a local redefinition of the loop operator, since it is linear in R, so the constant in R and epsilon piece is universal. This redefinition also eliminates the linear divergences in (12) and (13).

3. The connection to the Renyi entropy seems particularly nice. I would appreciate if the authors (perhaps in a follow-up) would explain the details of this connection.

Requested changes

1. If the authors wish, they can choose to address my questions and comments in the paper.

---

## Round 1 · Referee Report · Anonymous · 2021-4-5

Strengths

The authors identify the universal scaling behavior of the disorder operator in various quantum phase transitions in 2+1d.

Weaknesses

In most of the examples (including the ones in Section II and in Section III.A), it is not clear to me the role of the $\mathbb{Z}_N$ one-form or zero-form global symmetry in the microscopic systems. In these examples, the expectation value of the disorder operator, which is the main objective of this paper, is completely determined by the emergent $U(1)$ global symmetry data (which include $\sigma$) in the low energy CFT.

Report

The authors compute the expectation value of the disorder operator. The authors identify an interesting subleading log term when there is a corner, and show that the coefficient is determined by the current two-point function. This is an interesting result that deserves to be published.

Requested changes

1. The authors should clarify the role played by the $\mathbb{Z}_N$ zero-form and one-form global symmetry. See my comments in Weakness.

Concretely, I don't understand the significance of the $\cos(N\hat\theta_i)$ term in (1) (and the related $U(1)$-breaking, $\mathbb{Z}_N$-preserving terms in (2)). Since all the subsequent discussion and calculation assume that the $U(1)$ global symmetry reemerges in the low energy, why do we need to break it in the UV? It appears to me that the main result (which I find very interesting) has more to do with the $U(1)$ global symmetry at the critical point, and less to do with the $\mathbb{Z}_N$ zero-form or one-form global symmetry realized in the microscopic system. Indeed, in a closely related paper ref. 78, the authors there present similar results while keeping the $U(1)$ symmetry in the microscopic model.

2. In the discussion below (5), perhaps the authors can include the bootstrap reference 1912.03324, which reports the state of the art value of $C_J$ in the 2+1d O(2) CFT.

---

## Round 2 · Referee Report · Anonymous · 2021-5-21

Report

In the new footnote 83, the authors say that they are not certain about what the U(1) categorical symmetry is. I have a comment regarding this point.

It has long been known that if we gauge a $\mathbb{Z}_N$ 0-form symmetry of a 1+1d QFT $\cal T$, then the gauged theory ${\cal T}'$ also has a $\mathbb{Z}_N$ 0-form symmetry. The latter is known as the "quantum symmetry". If we gauge the quantum symmetry of ${\cal T}'$, we get back to $\cal T$. See "Quantum Symmetries of String Vacua" of C. Vafa in 1989.

This was further generalized in 1412.5148 to the following statement: if we gauge a $\mathbb{Z}_N$ $q$-form symmetry of a $d+1$-dimensional QFT $\cal T$, then the gauged theory ${\cal T}'$ has a $\mathbb{Z}_N$ $(d-q-1)$-form symmetry. If we gauge the $\mathbb{Z}_N$ $(d-q-1)$-form symmetry of ${\cal T}'$, we get back to $\cal T$.

In the past couple of years, this notion has been discussed under the name of "categorical symmetry" in the condensed matter literature.

What about continuous symmetries? It was famously known that if we gauge a $U(1)$ 0-form symmetry of a 2+1d QFT $\cal T$, then the gauged theory ${\cal T}'$ has a magnetic $U(1)$ 0-form symmetry. This is the $S$ of Witten's $SL(2,\mathbb{Z})$ action in hep-th/0307041. If we gauge the magnetic $U(1)$ 0-form symmetry of ${\cal T}'$, we get back to $\cal T$.

It appears to me that this is the proper notion of a "$U(1)$ categorical symmetry" in the condensed matter sense.

In fact, the explicit calculations in this paper are mostly about the Noether currents for the magnetic $U(1)$ symmetry. It therefore appears to me that this paper is really about the "$U(1)$ categorical symmetry" (in the sense I described above), rather than the "$\mathbb{Z}_N$ categorical symmetry".

I'll leave it to the authors' decision if they would like to further comment on this point.

---

## Round 2 · Referee Report · Anonymous · 2021-6-30

Strengths

Interesting and useful work

Weaknesses

Confusingly written

Report

A conservative point of view is that this paper is doing a simple and interesting thing, namely studying the universal bits of the behavior of disorder operators (with corners) at certain continuous phase transitions.

But they are describing what they are doing in a way that I find very confusing.

Specifically, I have to admit that I do not understand at all what the authors mean by "inexplicit dual symmetry". In fact, my understanding is that (for example in the Ising model example studied as example 1 in section IIA) it is only actually a symmetry at all when the system arises on the boundary of a higher dimensional system with topological order.
This apparently crucial point was made explicit in the earlier work of Wen and Ji (ref 16) but seems to not be mentioned at all here!
The authors know very well that duality does not preserve global data like the number of groundstates
(the 2d $Z_N$ Ising model in the broken phase has $N$ groundstates on the plane; this phase is dual to the deconfined phase of the $Z_N$ gauge theory, which has a unique groundstate on the plane).

No one will disagree that the correlations of the order parameter and disorder operator can be used to characterize these different phases, or that it is interesting to study their behavior at the critical point.

But since this obscure concept of "inexplicit dual symmetry" plays such a central role in the narrative structure of the present manuscript (particularly the abstract), I feel that it needs serious revision.

Furthermore, the authors' response to the very reasonable question 3 of Referee 1 I find completely mystifying and unhelpful. Part of the problem is that there are currently various competing definitions in the literature of the meaning of the term "categorical symmetry".

I also agree with another referee that what the authors call "ODO" is just a disorder operator. I don't understand the need to proliferate nomenclature here.

The work is clearly publishable, but I think the authors need to make more of an effort to explain their ideas clearly.

smaller comments:

-- Is it possible to understand the relation between the coefficient $b$ and the conductivity in some more general way? Why should there be such a connection?

-- Most of the calculation on page 3 of the manuscript is done in section VI of the famous review by Kogut:
https://journals.aps.org/rmp/abstract/10.1103/RevModPhys.51.659

-- I think the authors should at least make some comments about the difference between $ \langle \log^2 O\rangle $ and
the better-defined object $ \log \langle O \rangle $ in their interacting theories.

-- Corner contributions to Wilson (and 't Hooft) loop expectation values have been studied in gauge theory for several decades, for example under the name "cusp anomalous dimension".

-- The recent paper
https://arxiv.org/abs/2102.06223
gives a nice super-universal geometric explanation for the universal behavior of the angle-dependence of the corner contribution to various measures of fluctuations. I believe this result explains why the same function is seen in the entanglement entropy as in the behavior of Wilson loops.

-- Reference 9 (cited in the batch-reference in the first paragraph) is not about higher-form symmetry.

-- "Tt is known" should be "It is known"

-- "The angle dependence of the ODO is still give by Eq. 14"
should be "The angle dependence of the ODO is still given by Eq. 14"

Requested changes

Improve the presentation along the lines discussed in the report.

---

## Round 2 · Author Response

Response to report 1:

We want to thank referee 1 for very careful reading of our manuscript, and also very helpful comments.

Response to the questions:

  1. For Gaussian theory, one can simply take the exponential of <(log(O_C))^2> to reproduce <O_C>. However, for strongly interacting field theories, due to the lack of the Wick theorem, one can only say that <(log(O_C))^2> coincides with the second order expansion of the field operator of <O_C>. But one can alternatively view <(log(O_C))^2> as another quantity that diagnoses the phase transitions.

  2. About the numerical results in Figure 2, we simply mean that we performed a numerical integral of Eq.6 (for example) along the loop object. The numerical integral has no expansion of 1/L or epsilon, while the analytical evaluation in this paper extracts the dominant orders in the expansion of 1/L and epsilon.

  3. The referee raised very good question about orbifold. We will do our best to give a brief review of (our understanding of) categorical symmetry and ODO here. In recent years the concept of symmetries has been significantly generalized, and many concepts that were thought to be beyond the Landau’s paradigm (such as emergent photon, topological order), can now be interpreted in the language of spontaneous symmetry breaking (SSB) of generalized symmetries. The categorical symmetry is one step further of generalizing the notion of symmetry. For example, there are two 1d boundary states of the 2d toric code separated by an unavoidable phase transition. Neither boundary phase has ground state degeneracy, but they can still be described as SSB of two “Z2 symmetries”, but the physical ground state can be viewed as the orbifold of the corresponding Z2 symmetry. In fact in the first paper that introduced the categorical symmetry (Ref.16), the authors were specifically discussing the “symmetric sector” of the Ising model, which is the orbifold of the Z2 symmetry. The situation is similar to the “dual inexplicit symmetries” in higher dimensions, where the ground state is an orbifold of the dual symmetry. Many usual concepts of symmetry and symmetry breaking no longer apply here. This is why we introduced the notion of ODO to describe systems with a symmetry, or orbifolds of symmetry (using the referee’s language). As long as we use the correct notion, we believe the concept of categorical symmetry is indeed well-defined: there is still an unavoidable phase transition between the disordered and ordered phases, even if we only consider the orbifold of a symmetry, and the ODO is the quantity to diagnose the two phases.

Response to the minor comments:

  1. Yes, the “ODO" in the manuscript indeed corresponds to the disorder operator in the case of Ising model. The concept of ODO was developed for more general cases, including cases with higher symmetries, or subsystem symmetries as well. We have added a footnote to explain that for the simple Ising model the ODO is the same as the disorder operator.

  2. Yes we agree the linear divergences in all ODOs studied here can be eliminated, and the universal log contribution is the most interesting object to study.

  3. Indeed, the connection to the Renyi entropy is very interesting. We are in the process of further study, and we would prefer to leave it for future work.

Response to report 2:

We really appreciate the helpful comments and suggestions from referee 2.

Response to the questions:

1, About Z_N v.s. U(1)

This is a very good question. Indeed most of the calculations can go through with just U(1) symmetry, and we are aware that Ref. 78 (now Ref.79) studied the case that kept the U(1) symmetry. However, we have concerns about the lattice definition of categorical symmetry with respect to an original U(1) symmetry. When the original symmetry is Z_N, the ODO of the dual 1-form symmetry is a line/loop object defined on the lattice, whose behavior diagnoses the order and disordered phase of Z_N; but for the cases with U(1) symmetry, we know that the “disorder” is driven by vortices, rather than a line object. This may be just an unnecessary philosophical concern, but we feel the physical picture is most clear for the Z_N categorical symmetries. Generalization of categorical symmetries to continuous symmetry is possible, but we prefer to leave this to more careful future study. We have added a footnote to explain this point.

  1. We are very grateful to the referee for bringing the reference to our attention, we have added it to our reference list.

---

## Round 2 · List of Changes

1, we added a footnote to explain the relation to the disorder operator defined previously, as suggested by referee 1;

2, we added a footnote to explain the relation between Z_N and U(1) symmetries, raised by the referee 2. In general, generalization of categorical symmetries to continuous symmetry is possible, but we prefer to leave this to more careful future study.

2, references suggested by the referees were added (Ref.48 and Ref.80).

---

## Round 3 · Author Response

Dear editor,

We want to thank the referee for all the suggestions. Here is our response to the new report:

1, We have added a new appendix to clarify all the notions used in the manuscript. We believe it should be clear now. In short, we listed two qualities that a standard “explicit symmetry” should satisfy. An “explicit symmetry” and the dual “inexplicit symmetry” used in this manuscript both satisfy quality (1), which corresponds to the conservation law of excitations, but the dual “inexplicit symmetry” does not satisfy quality (2). Both symmetries can be made explicit (meaning they both satisfy (1) and (2)) by embedding the system to the boundary of higher dimensional topological order (like Ref.16, now Ref.15). In this perspective the conservation laws associated with these two “symmetries” arise from the fusion rules of the anyons in the bulk. But we would like to emphasize that the quantities we are interested in can be defined and computed without the bulk.

2, Ref.16 (now Ref.15) first introduced a new phrase “patch operator” as a generalization of the disorder operator. Indeed, we also feel a new phrase is necessary for more general cases, because the phrase “disorder” operator implies that when it condenses, the original symmetry would be restored or the system should enter a disordered phase of the original symmetry. But in some cases that involve higher form symmetries both the symmetry and the dual symmetry can be spontaneously broken simultaneously, namely both the explicit symmetry and its dual inexplicit symmetry can enter the ordered phase simultaneously under proper generalization. But the phrase “patch operator” is also not satisfactory: it is not clear about the function and purpose of this quantity. And for situations that involve more exotic subsystem symmetries, the desired quantity is not defined on a simple patch of the system. Hence together with one of the authors of Ref.16 (now 15), we proposed the phrase “order diagnosis operator” in our previous work.

We would also like to mention that, overlap between different concepts/notions/phrases, though not ideal, is very common at an early stage of the research on a subject. For example, in the last 10 years the integer quantum Hall state was categorized as either “short range entangled state” (in the sense that it does not have topological entanglement entropy) or “long range entangled state” (in the sense that it cannot be connected to simple product state through finite depth quantum circuit) by different researchers, and later it was also categorized as “invertible topological order”. In more recent literature it seems researchers are converging to the phrase “invertible topological order”, as this phrase more accurately captures the essence of the concept. Also, the phrase “symmetry protected topological states” was also called “symmetry protected trivial states” in some literature (in fact both phrases were used by the same researcher). Time will tell what phrase is the most appropriate; and before that, different authors should be allowed to use the phrase that they consider most appropriate. We do believe the phrase “order diagnosis operator” properly captures the purpose of the series of desired quantities. And we believe the usage of nomenclature is self-consistent in the current version of our work.

3, we only learned about the reference mentioned by the referee (2102.06223) after our work was posted (the reference was posted one month after our work). Indeed, this reference made a general analysis of correlation functions integrated on an area, and a corner contribution is universal. We have added a footnote and cited the main result of this reference.

4, Regarding the question “Is it possible to understand the relation between the coefficient b and the conductivity in some more general way”, we think the procedure we used in the paper is already quite general: the computation of b is reduced to a density-density correlation, which in a system with Lorentz invariance in the IR is proportional to the current-current correlation, which is proportional to the conductivity.

5, in the previous version of our manuscript we actually have cited two references from the high energy physics literature that computed the cusp contribution to the Wilson loop of gauge field. We have now cited these two references in the introduction section, together with the review article mentioned by the referee.

6, We believe we did mention why we choose <(log O)^2> over log <O> in the previous version of our manuscript: for an interacting theory, log <O> can only be evaluated by expanding the quantity as a polynomial of fields, and the second order can be evaluated conveniently. So far we have not proven whether higher order in this expansions would change the corner dependence or not. The definition of <(log O)^2> was further explained in one of the footnotes.

7, the typos mentioned by the referee were corrected. And Ref.9 is removed.

---

## Round 3 · List of Changes

The main change is an appendix meant to clarify all the rudimentary notions used in this manuscript. Other changes are marked blue in the text.

---

## Editorial Decision

published